# Protein design using structure-based residue preferences

**David Ding** [1] ✉, **Ada Y. Shaw**[2], **Sam Sinai**[3], **Nathan Rollins** [4], **Noam Prywes** [1], **David F. Savage** [1,5,6], **Michael T. Laub** [7,8] **& Debora S. Marks** [2] ✉

Recent developments in protein design rely on large neural networks with up to 100s of millions of parameters, yet it is unclear which residue dependencies are critical for determining protein function. Here, we show that amino acid preferences at individual residues—without accounting for mutation interactions—explain much and sometimes virtually all of the combinatorial mutation effects across 8 datasets ($R^2$ ~ 78-98%). Hence, few observations (~100 times the number of mutated residues) enable accurate prediction of held-out variant effects (Pearson r > 0.80). We hypothesized that the local structural contexts around a residue could be sufficient to predict mutation preferences, and develop an unsupervised approach termed CoVES (<u>Co</u>mbinatorial <u>V</u>ariant <u>E</u>ffects from <u>S</u>tructure). Our results suggest that CoVES outperforms not just model-free methods but also similarly to complex models for creating functional and diverse protein variants. CoVES offers an effective alternative to complicated models for identifying functional protein mutations.

A key question in molecular evolution and protein engineering is how multiple mutations combine to impact both function and future mutational trajectories. The possible mutational trajectories of a given protein can be limited, if, for example, the negative effect of a single substitution can only be tolerated in the presence of another enabling mutation[1–3]. Conceptually, such specific dependencies between mutations cause 'rugged' fitness landscapes, in which natural or experimental selection for fitness-increasing mutations does not necessarily result in optimally functioning proteins[4]. On the other hand, if multiple mutations combine without specific dependencies between each other, the sequence-fitness function will result in a simple monotonic function on which selection can act more efficiently. Similarly, knowing such specific dependencies is critical in determining combinations of mutations for the design of protein therapeutics with desired functions.

While recent advances in modeling protein function in response to mutations have focused on increasing the capacity of models to fit more complex fitness landscapes[5–10], it remains unclear how complicated biological protein fitness landscapes are. The explicit number of required specific dependencies, i.e., epistatic terms, grows combinatorially with the order of interactions considered $\left(20^k \binom{L}{k} \frac{1}{k!}\right)$ terms of order k and protein length L). For example, for a protein of length 100, there are 2000 first-order site-wise terms, but ~1 million second-order pair-wise interaction terms and ~200 million third-order interaction terms. Indeed, sequence models that are able to capture dependencies between residues show increased performance in predicting observed variant effects[5]. Recent efforts to predict combinatorial protein variant effects have adapted large neural networks from other domains, some with more than a billion parameters, to implicitly capture such dependencies when trained on protein sequences[6,10]. Such models can require not just vast amounts of data, but also costly compute resources, non-trivial tuning of hyperparameters and time for training. Overparameterized models are also prone to overfitting, notoriously hard to interpret, and hence can produce false positive predictions. For most proteins it is not clear how many and which dependencies are actually required for accurately predicting

[1]Innovative Genomics Institute, University of California, Berkeley, CA 94720, USA. [2]Department of Systems Biology, Harvard Medical School, Boston, MA 02115, USA. [3]Dyno Therapeutics, Watertown, MA 02472, USA. [4]Seismic Therapeutics, Lab Central, Cambridge, MA 02142, USA. [5]Department of Molecular and Cell Biology, University of California, Berkeley, CA 94720, USA. [6]Howard Hughes Medical Institute, University of California, Berkeley, CA 94720, USA. [7]Department of Biology, Massachusetts Institute of Technology, Cambridge, MA 02139, USA. [8]Howard Hughes Medical Institute, Massachusetts Institute of Technology, Cambridge, MA 02139, USA. ✉e-mail: davidding@berkeley.edu; debbie@hms.harvard.edu

combinatorial protein variant effects. The complexity of the biological fitness landscape, i.e. how many dependencies need to be considered, will directly determine the required capacity of any model to approximate such fitness functions well.

To determine the importance of epistasis in protein fitness landscapes, we examined combinatorial variant effects across 6 proteins (using 8 separately collected datasets, including one generated here). We found that the measured combinatorial variant effects across these proteins can be well explained ($R^2$-0.78–0.98) by a function that only considers the 20*N residue mutation preferences, where N indicates the number of mutated positions, passed through a global nonlinearity[11], without considering specific dependencies between mutations. We observed that a small number of observations (5-fold oversampling the number of residue-wise mutation preference parameters, and, in one dataset, as few as 100–200 observations) is sufficient to enable high predictive accuracy on held-out combinatorial variant effects (Pearson r > 0.8), outperforming any unsupervised methods for predicting variant effects.

The high general applicability and performance of such per-residue mutation preference models led us to devise an unsupervised strategy that we term CoVES—for 'Combinatorial Variant Effects from Structure'. CoVES designs functional and diverse protein variants without experimental variant effect measurements by inferring the required residue-wise mutation preferences using an equivariant graph neural model which takes the structural context surrounding the residue as input (Fig. 1a). CoVES performs similar to complicated, high-capacity neural models, which model long-range specific dependencies between primary sequence positions, in functional and diverse sequence design, when evaluated with near perfect surrogate fitness functions for two proteins examined here (Fig. 1b, c).

Collectively, our findings systematically illuminate the simplicity of local protein fitness landscapes in many cases, suggest minimum experimental measurement rules for enabling supervised protein variant design, and present a biologically-rooted unsupervised strategy, CoVES, to design combinatorial protein variants with negligible time and compute requirements in the absence of data. The performance of such microenvironment-only approaches suggests that structural context could be sufficient for protein design.

## Results

### High-throughput in vivo measurement of combinatorial variant effects in ParD3

To understand how individual substitutions in proteins impact each other, we first focused on measuring the effect of combinatorial variants in the *Mesorhizobium opportunistum* antitoxin ParD3, which was randomly mutated at 10 positions and assayed for neutralization of its cognate toxin ParE3 (Fig. 2a). This set of 10 residues was chosen as they directly contact the toxin, and are among the top co-evolving residue pairs across the toxin-antitoxin interface[11]. A library of cells containing different antitoxin variants was grown in bulk; only those cells containing antitoxin variants that neutralize the toxin can proliferate. The change in frequency of each antitoxin variant was followed via high-throughput sequencing over time (Fig. 2b). These variant effect measurements correlate with orthogonal growth rate measurements and have high reproducibility between biological replicates[11–13]. We used this assay to measure the growth rate effects of 7923 combinatorial amino acid variants spanning the 10 binding residue positions and 2615 truncated antitoxins with stop codons (Fig. 2a, Supplementary Fig. 1). We calculated the growth rate of each variant as the normalized log read ratio before and after selection, and found good separation

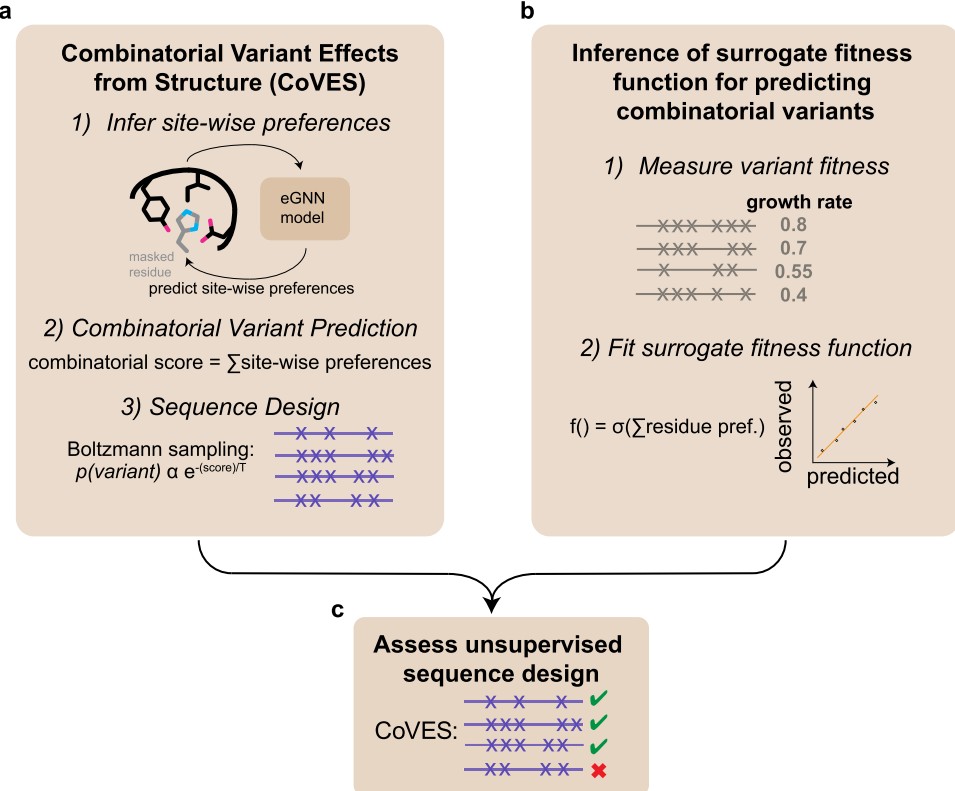

**Fig. 1 | Design of protein sequences using structural information alone and assessment of designed sequences using surrogate fitness functions trained on experimental observations. a** Structural environment predicts residue mutation preferences and can be used for designing combinatorial protein variants. **b** Learning supervised fitness functions from experimental high-throughput variant measurements. The functional form of the fitness function, f(.), can be learned by fitting to observed data, and enables predicting the function of unobserved sequence variants. **c** Assessment of designed sequences with the surrogate fitness function enables comparing different sequence design strategies. Source data are provided as a Source Data file.

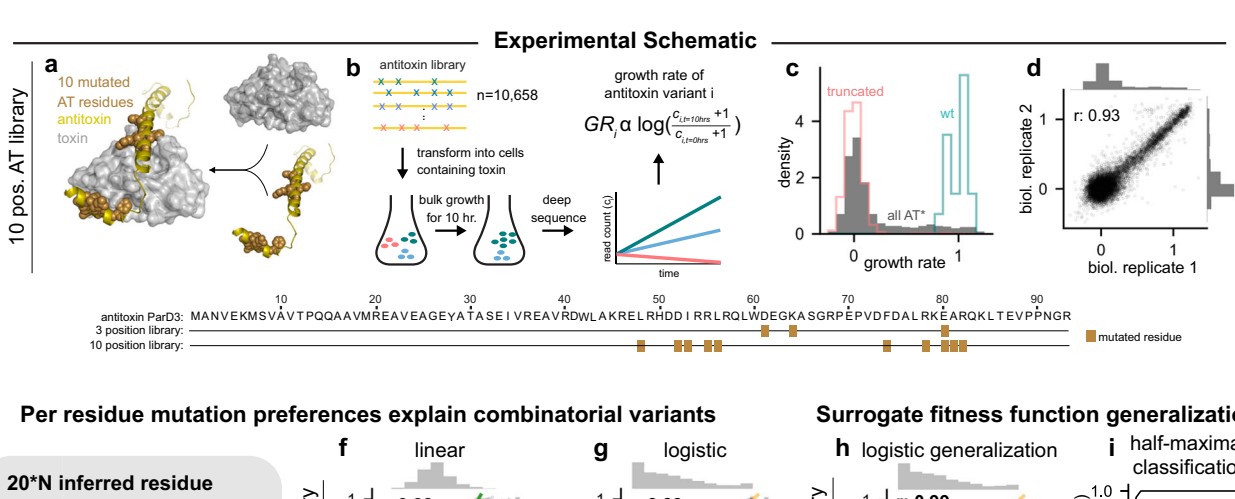

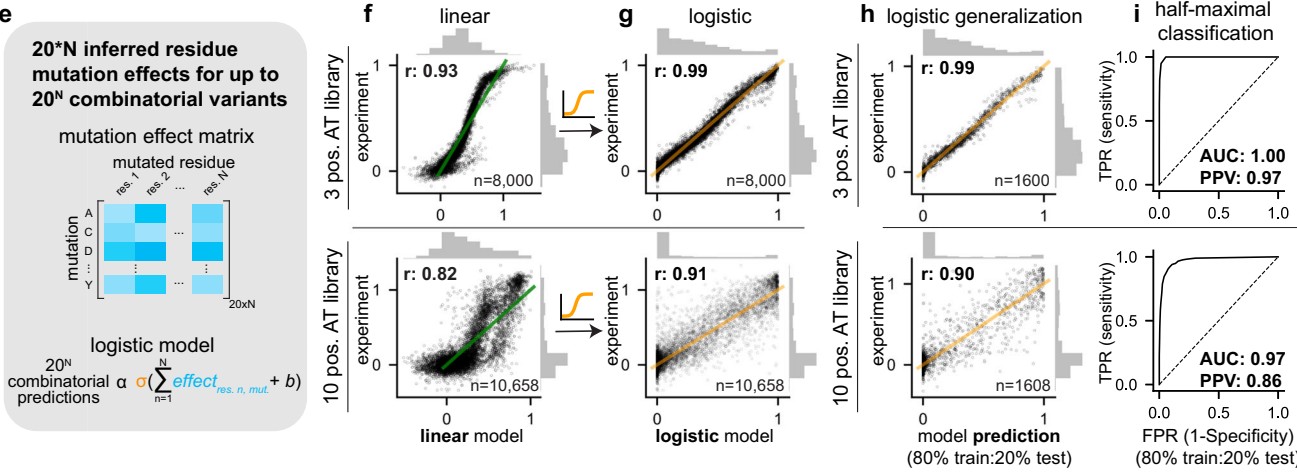

**Fig. 2 | Residue amino acid preferences can explain combinatorial mutation effects for multiple proteins and enable predicting the function of unobserved combinatorial variants. a–d** Ten binding residues (AT: L48, D52, I53, R55, L56, F74, R78, E80, A81, R82; $n = 10,658$) of the antitoxin ParD3 were randomized (shown space-filled on PDB ID:5CEG, panel **a**) and highlighted on the antitoxin sequence (bottom), transformed into cells containing wild-type toxin ParE3, and the growth of individual antitoxin variants followed by high-throughput sequencing over two timepoints to calculate the normalized log read ratio (growth rate, GR) for each variant (**b**). Antitoxin variants that are able to bind and neutralize the toxin will show higher growth rates. The distribution of measured growth rate values for all antitoxin variants, wild-type antitoxin, and truncated antitoxins is shown (**c**). The reproducibility of growth rate values between two biological replicates (**d**). **e–g** The

logistic regression model learns the 20*N per residue mutation effects (**e**) before passing through a sigmoid function (orange) to predict $20^N$ combinatorial variants. Logistic regression (**g**) fits the observed combinatorial variants better than linear regression does (**f**). Top row shows fit to 3 position randomized antitoxin library, bottom row shows 10 position randomized antitoxin library. The logistic regression model enables predicting held-out 20% of the random combinatorial variants (**h**), and enables classification of half-maximal neutralization (**i**). **j** Total variance explained and held-out correlation of site-wise logistic regression model across 8 combinatorial variant datasets. **k** A subset of the observed combinatorial variants is sufficient to infer the site-wise preference parameters to explain the remaining held-out combinatorial variant effects. Source data are provided as a Source Data file.

between truncated and wild-type antitoxin variant effects (Fig. 2c). This assay exhibited high reproducibility between separate biological replicates (Pearson r: 0.93, Fig. 2d), but we note that nonfunctional antitoxin variants with growth rate values below -0.2 could not be resolved. As expected, the distribution of fitness effects shifted towards loss of function as more substitutions were introduced, with the majority of variants achieving at least half-maximal toxin

neutralization when fewer than three substitutions were present (Supplementary Fig. 1a). With more than three substitutions, the fraction of functional variants decreases faster than exponentially (Supplementary Fig. 1b). Using this maximum likelihood estimate of the fraction of functional variants and considering the total number of possible mutations at each mutation distance, we estimated that there are ~3 ×$10^{10}$ combinatorial variants – out of a possible $20^{10}$-$10^{13}$

combinatorial variants among these 10 residues—that achieve half-maximal neutralization when these ten positions are mutated (Supplementary Fig. 1c).

## A simple, logistic regression model explains combinatorial variant effects in the antitoxin ParD3

We sought to understand how well observed combinatorial variant effects could be explained by considering only per position amino acid preferences, without specific interaction terms between mutated residues. To do so, we examined two datasets that measured combinatorial variant effects in the ParD3 antitoxin for cognate toxin neutralization (Fig. 2a): The 10 position library dataset generated in this study, as well as a 3-position, combinatorially exhaustive library from Ding et al.[11] (8000 combinatorial variants at positions D61, K64 and E80, with Pearson r = 0.98 between biological replicates). We first fit a linear regression model to each of these datasets, with per residue mutation preferences as predictors, but without specific interaction terms between residues. This model estimates a latent residue preference effect for each individual amino acid variant at each position, and sums these effects to predict combinatorial variants. This approach performed well (3 position dataset Pearson r: 0.93, 10-position dataset: 0.82), but shows markedly biased residuals between fitted and observed growth rate effects (Fig. 2f, Supplementary Fig. 2a). We also examined whether simply adding the observed single variant effects are predictive of combinatorial variant effects (Supplementary Fig. 2b). The predicted and observed growth rate effects correlate poorly (3-position dataset Pearson r: 0.48, 10-position dataset: 0.57).

We then fit a nonlinear, site-wise model (i.e., a logistic regression model with only per residue amino acid preferences as predictors, Fig. 2e, g; Supplementary Fig. 2c) to each dataset. This model infers mutational preferences for each residue using the observed combinatorial variants, sums the relevant inferred preference parameter for each mutation in a particular combination of variants, and then passes this sum through a nonlinear sigmoid function to predict the effect of a combinatorial variant (Fig. 2e, Supplementary Fig. 2c). This sigmoid function could account for nonlinearities arising from growth rate saturation or binding curves[11]. This model enabled significantly better, and in the case of the 3 position library, almost perfect fit to the observed combinatorial variant effects (Fig. 2g; 3-position dataset: 0.98 Pearson r or 98% explained variance $R^2$; 10-position dataset: 0.91 Pearson r or 83% explained variance $R^2$), without biased residuals between fitted and observed combinatorial variant effects.

Importantly, this non-linear, mutation preference model used a linearly increasing number of parameters (20*N preference parameters + 1 bias parameter, where N indicates the number of mutated positions) to explain an exploding number of combinatorial variants ($20^N$) (Fig. 2e), i.e. the 8000 (=$20^3$) observed combinatorial variant effects in the antitoxin 3 position library were explained by a logistic regression model with 61 (=20*3 residue mutation preference + 1 bias) parameters.

## Per residue mutation effect models can explain combinatorial variant effects across proteins

We wondered whether the performance of such non-epistatic, per residue mutation effect models for fitting observed combinatorial variants generalizes to other protein functions and folds. Indeed, the explanatory power of such models have been described in previous cases[11,14–17]. To systematically examine the applicability of such models, we identified 5 existing variant datasets that contain combinatorial variant effect measurements, including variants in the IgG-binding domain of protein G (GB1[18]), poly(A)-binding protein (PABP[19]), the SH3-domain of the human growth factor receptor-bound protein 2 (GRB2[20]), model-designed combinatorial variant effects in adeno-associated virus (AAV[21]), random error-prone PCR mutations in green fluorescent protein (GFP_SAR[16]), and focused mutations at 13 special residues in green fluorescent protein that enable switching phenotype (GFP_POE[17]). We then fit a per residue, nonlinear regression model to these variant datasets (Table 1, Supplementary Fig. 3a, b). Even though these assays have varying experimental noise, we observe that similar to the toxin-antitoxin case, a nonlinear, non-epistatic mutation preference model could explain 78–95% of the observed mutant effects across these datasets (Fig. 2j, Table 1). A linear regression model with the same number of mutation preference parameters, but lacking the non-linear transformation function performed less well in all cases (Supplementary Fig. 3a). Strikingly, even for the GFP_POE dataset, in which only phenotype switching residues are mutated and higher order epistasis has been detected before[17], such a per-residue effects model can still explain 94% of the observed variance. These results suggest that summed per residue mutation preferences passed through a nonlinear transformation, without considering specific dependencies between mutations, are a powerful tool for explaining combinatorial variant effects across multiple proteins.

## Few observations are sufficient to predict unobserved combinatorial mutation effects

Because we were able to explain a large number of combinatorial variant effects using few mutation preference parameters, we next asked whether a smaller subset of random observations would be sufficient to infer these preferences. We repeated inference of the mutation preferences of the nonlinear, residue mutation effect regression model using smaller, random subsets of the total observed dataset in each case, and evaluated these models for predicting the fitness effect of combinatorial variants in the remaining dataset. Indeed, models trained on a much smaller number of random observations were sufficient to explain the unobserved combinatorial mutation effects well (Supplementary Fig. 4). Strikingly, for the ParD3 antitoxin 3 position library, 100 or 200 random combinatorial variant effect measurements were sufficient to achieve almost perfect correlation (Pearson r = 0.98) between the observed and predicted held-out combinatorial variants (Supplementary Fig. 4a top).

**Table 1 | Summary statistics for examined datasets**

| Protein | Logistic Fit ($R^2$) | Logistic generalization (90–10% test Pearson r) | # observed mutants | # model parameters | # obs. / # params | # mut. sites | Experimental reproducibility | Mutation range | Average # mutations | dispersed through structure? |
|---|---|---|---|---|---|---|---|---|---|---|
| AT 3pos. | 98% | 0.99 | 8000 | 61 | 131.15 | 3 | 0.99 | 1–3 | 2.85 | No |
| AT 10pos. | 83% | 0.90 | 10,658 | 201 | 53.02 | 10 | 0.93 | 1–10 | 5.38 | No |
| GB1 | 95% | 0.97 | 536,962 | 1158 | 463.70 | 52 | 0.997 | 1–2 | 2.00 | Yes |
| AAV | 78% | 0.87 | 42,328 | 591 | 71.62 | 28 | 0.89 | 1–21 | 4.73 | No |
| GRB2 | 89% | 0.94 | 63,367 | 1179 | 53.75 | 55 | 0.92 | 1–2 | 1.98 | Yes |
| PABP | 89% | 0.92 | 36,521 | 1142 | 31.98 | 74 | n/a | 1–2 | 1.97 | Yes |
| GFP_SAR | 89% | 0.93 | 51,714 | 5001 | 10.34 | 233 | n/a | 1–15 | 3.88 | Yes |
| GFP_POE | 94% | 0.96 | 8192 | 16 | 512.00 | 13 | n/a | 1–13 | 6.50 | No |

We hypothesized that the required training set size is dependent on the number of randomized positions, and the resulting number of mutation preference parameters, which in turn differ as a function of the number of mutated positions in each dataset. We therefore visualized the performance of models trained on a subset of datapoints relative to the number of mutation preference parameters in each model. This demonstrated that a mere oversampling of 5-fold in terms of training set size relative to the number of parameters in all 7 datasets was sufficient to enable prediction of unobserved combinatorial variants with a Pearson correlation coefficient r > 0.8, outperforming state-of-the-art unsupervised models[9] (Fig. 2k).

These results indicate that measuring only 5*N*20 combinatorial variant effects are sufficient to fit the N*20 mutation preference parameters of this non-epistatic, nonlinear model at N residues in order to predict observed combinatorial variants effects across 7 protein variant effect datasets.

### Mutation preference non-linear regression can serve as a 'surrogate fitness function' to predict the effect of unobserved combinatorial variants

We also tried to estimate how well such per residue, nonlinear regression models can predict unobserved variant effects. For example, how well can this model trained on ~8000 antitoxin variants at ten randomized positions be used to predict the possible $20^{10}$ possible combinatorial variants? To estimate the generalization error beyond the observed variants, we inferred parameters from a random 80% subset of the total observed combinatorial variants, and tested the predictive accuracy of the model on the held-out remaining 20% of observed variants. The correlation between predicted and measured held-out test variants was high (Fig. 2h, Pearson r: 0.99 for the 3 position library and Pearson r: 0.90 for the 10 position library). This model had high performance for classifying held-out combinatorial variants that achieve at least half-maximal fitness (3 position antitoxin library AUROC: 1.00, positive predictive value (PPV): 0.97; 10 position antitoxin library AUROC: 0.97, PPV: 0.86). These findings suggest that the non-epistatic logistic model can be used as an 'surrogate fitness function' to predict the effect of unobserved combinatorial variants among these ten antitoxin positions.

### A per residue, unsupervised method, CoVES, trained on the structural microenvironments around each residue, can recover mutation preferences and score observed combinatorial variant effects across datasets

Is it possible to recover the required per residue mutation preferences and predict combinatorial variant effects without measurements? We observed that the inferred per residue mutation preferences in the 3 position antitoxin library can be altered by mutations in contacting residues in the toxin (Supplementary Fig. 5). Given this and the performance of non-epistatic models to capture combinatorial variants, we hypothesized that the three-dimensional structural environment around a particular amino acid could be sufficient to learn the per residue mutation preferences of interest and thereby predict combinatorial variant effects.

To test whether the structural microenvironment is sufficient to learn per residue mutation preferences and predict combinatorial variants, we developed a strategy—which we call CoVES for 'Combinatorial Variant Effects from Structure' (Fig. 3a)—by training a graph neural network[22] that learns rotation equivariant transformations to predict the identity of a masked amino acid from its atomic level 3D environment across non-redundant structures from the Protein Database (PDB) (test accuracy = 52.8%, compared to ~5% for random guess, training and testing dataset from Townshend et al.[23]). We then predicted the amino acid preference at each of the mutated positions across the combinatorial variant datasets given their respective

structural environments (Fig. 3a). Indeed, this graph neural network was able to recover the per residue mutation preferences which explained the observed combinatorial variant effects in the antitoxin (Supplementary Fig. 6, average pearson r across 3-position antitoxin library residues: 0.80, average pearson r across 10-position antitoxin library residues: 0.57). To predict combinatorial variant effects, we summed the log probability of amino acid preferences at each mutated position to predict the effects of combinatorial variants. This strategy was able to predict observed combinatorial mutation effects similarly well to other state-of-the-art unsupervised variant effect predictors across 5 variant effect datasets with proteins containing structural information (Fig. 3b, Supplementary Fig. 7).

### CoVES trained on structural microenvironments can generate diverse and functional antitoxin sequences

We assessed whether the unsupervised, structural context-only CoVES approach was able to design diverse and functional sequences at these 10 antitoxin positions (Fig. 3a). This task differs from combinatorial variant score prediction as emphasis is placed on sampling truly high functioning mutations, rather than predicting the entire range of mutation effects. Given the amino acid preference scores learned from each mutated residue's structural environment, we used a Boltzmann energy function to sample sequences (n = 500) at various temperatures, t, to control the diversity of the generated sequences.

To test the ability of CoVES for diverse and functional sequence design, we synthesized and tested 11 randomly sampled antitoxin variants (t = 1.5) for their ability to neutralize the toxin. Six of these antitoxin variants (6/11–55%), containing 2–7 mutations from wild-type antitoxin, were able to neutralize the toxin in a plate-based serial dilution growth assay (Fig. 3c). We also tested the functionality of CoVES-sampled antitoxin variants at a given mutation distance (3, 5, 6, 8 and 9) from wild-type with the best score of the surrogate fitness function. Indeed, 4 out of 5 antitoxin variants (80%) are able to neutralize the toxin (Fig. 3c). These results demonstrate that CoVES can design functional and diverse antitoxin variants.

How well does this per residue, unsupervised strategy compare to ones that can consider arbitrary specific dependencies between residues? To address this question, we compared generated sequences from CoVES against sequences designed from two models that can consider the sequence context in an autoregressive manner (proteinMPNN from Dauparas et al.[24], Ingraham et al.[25]; ESM-IF from Hsu et al.[8]). The high predictive accuracy of the supervised surrogate fitness function to predict half-maximal neutralization of held-out combinatorial variants in the experimental 10 position antitoxin library (AUROC = 0.97, Fig. 2i) gives us an opportunity to assess the function of designed sequences that are not observed in our high-throughput assay. At temperature t = 1.5, CoVES generated 91 unique sequences of which ~70% are predicted by the surrogate fitness function to achieve half-maximal fitness in the bulk growth rate assay (Fig. 3d). These sequences showed an average of 6.7 substitutions between samples and 5.6 with respect to the wild-type antitoxin (Fig. 3e). Inspecting the generated sequences indicated that multiple different substitutions can occur at each position (Fig. 3e).

Because sampling temperatures do not have a 1:1 correspondence between models and sampling algorithms, we sampled sequences for each model at a range of temperatures and assessed each set of sampled sequences for the fraction that is predicted to be functional (ie. at least half-maximally neutralizing) by the 'surrogate' fitness function, as well as their diversity. We found that CoVES outperformed both of these models for generating putatively functional sequences that diverge from each other and with respect to the wild-type antitoxin sequence (Fig. 3f, Supplementary Fig. 8). All of these unsupervised models, which consider structural features, also outperformed simulated annealing sampling from EvCouplings (Fig. 3f yellow), which

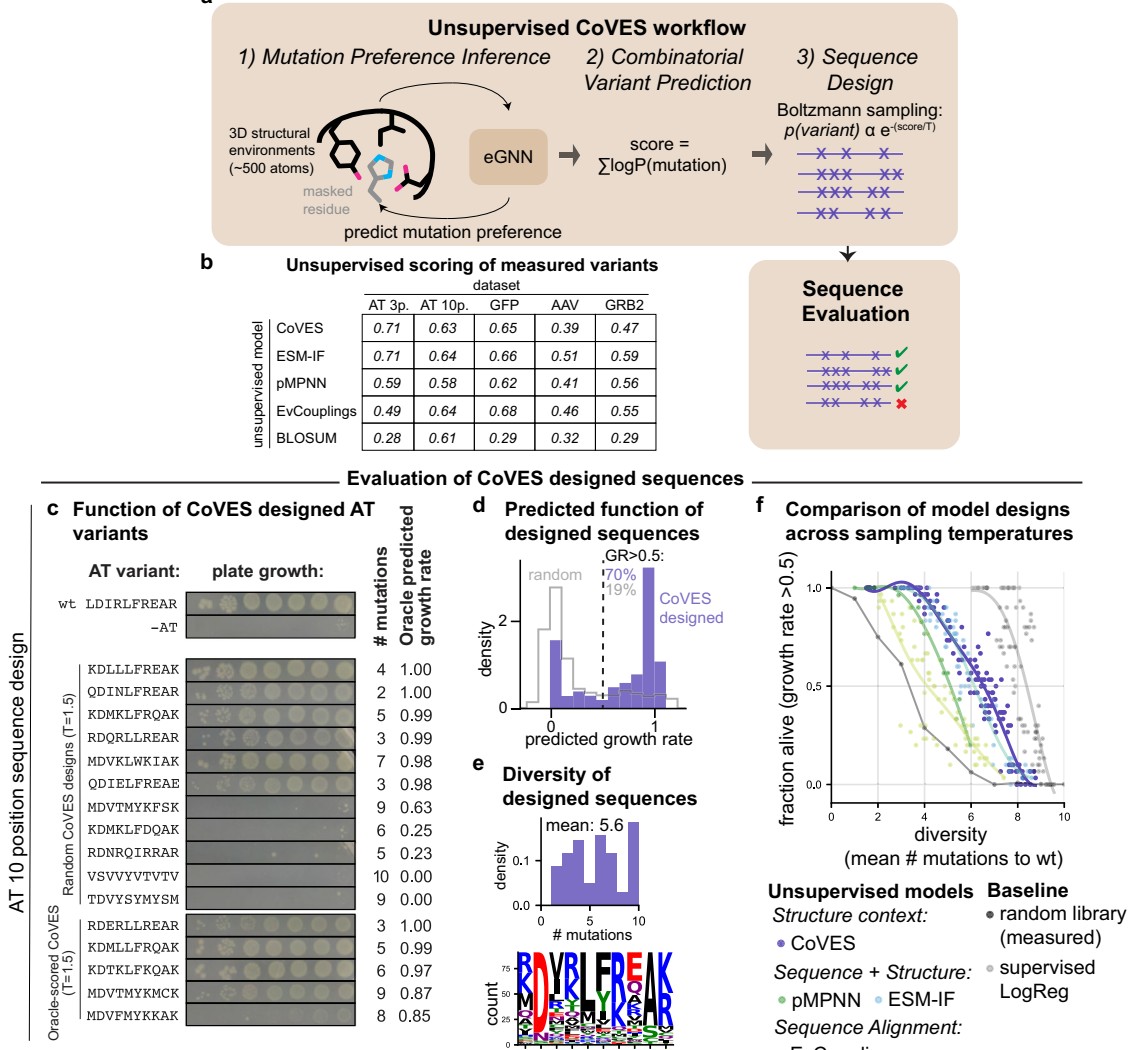

**Fig. 3 | CoVES, an unsupervised approach to learn residue mutation preferences from structural microenvironments, can predict variant effects and generatively design functional and diverse sequences. a** Schematic of the CoVES workflow: First, an equivariant graph neural network[23,31] is used to predict amino acid preferences from the structural environment around a particular residue. The mutation preference for each residue are converted into log probabilities, and summed to predict combinatorial variant effects. Finally, these scores can be used to design combinatorial variants at the desired residue positions by sampling with the Boltzmann energy function. **b** Spearman correlation coefficients between unsupervised model scores and observed combinatorial variant effects. **c** Serial dilution growth assay of CoVES designed antitoxin variants in the presence of wild-type toxin. **d**, **e** CoVES designed antitoxin sequences evaluated for their predicted functionality and diversity. Generated sequences ($n = 91$ unique sequences,

temperature = 1.5) from CoVES are evaluated for their predicted growth rate effects using the supervised surrogate fitness function (**d**), and their mutation number with respect to the wild-type antitoxin (**e**). **f** Comparison of generated sequences using CoVES vs. other state-of-the-art sequence design models in terms of the fraction of generated antitoxin sequences predicted to be functional and their average number of mutations as the sampling temperature is varied. Each dot represents a collection of sampled sequences summarized by their average number of mutations and their predicted fraction of functional sequences. Random library measurements indicated in dark gray, and energy-based Boltzmann sampling from the supervised, per residue logistic regression model trained on the observed variant data indicated in light gray. Lines represent polynomial fits. Source data are provided as a Source Data file.

learns site-wise and pair-wise mutation preferences from the natural sequence alignment, and outperformed model-free random sampling of sequences observed in the combinatorial variant library (Fig. 3f dark gray). We note that simple Boltzmann sampling using the scores from the supervised 'surrogate fitness' model, i.e., the logistic regressor trained on observed combinatorial variant effects to learn amino acid preferences, outperforms all unsupervised strategies outlined above (Fig. 3f, light gray).

### CoVES can generate diverse and functional GFP sequences

We next asked whether CoVES sampling can generate functional and diverse sequences in a different protein fold and function, as well as

across more than fixed 10 residues. We chose to focus on designing GFP sequences, for which an accurate surrogate fitness function could be trained using random mutation data from ref. 16, to predict 10% held-out combinatorial variant datasets with up to 15 mutations spread over 233 GFP residues (Pearson r: 0.93, Fig. 4a). Almost all combinatorial held-out variants that are predicted by the surrogate fitness function to achieve at least half-maximal fluorescence do achieve this threshold in the experiment (Fig. 4b, c, positive predictive value = 0.99, AUROC for half-maximal fluorescence classification: 0.99). We note that this surrogate fitness function is expected to perform well only for in-distribution generalization, i.e. for predicting combinatorial variants consisting of individual mutations that are observed during

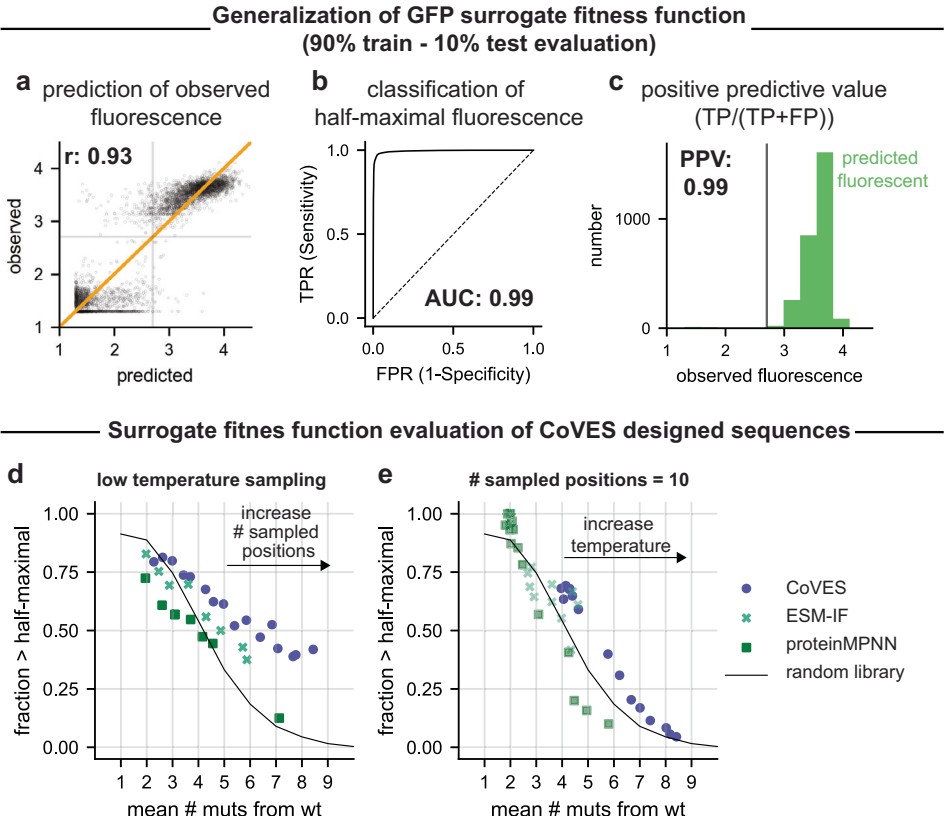

**Fig. 4 | Evaluation of function and diversity of CoVES-designed GFP variants.**
**a**–**c** GFP surrogate fitness function performance evaluated on held-out 10% test data, when trained on 90% random training data. **a** The correlation between predicted and observed fluorescence values is shown. The Pearson correlation coefficient, r, is indicated. **b** The true positive rate (TPR) vs. false positive rate (FPR) curve of half-maximal fluorescence classification is shown. **c** The positive predictive value, ie. fraction of truly above half-maximal fluorescing variants among the total number of predicted above half-maximal fluorescing GFP variants is shown for the held-out 10% training data. **d**, **e** The diversity and function of sampled sequences from unsupervised models is shown. Each dot represents a set of sampled

sequences at a given temperature and a fixed number of positions that are randomized, summarized by the fraction of sequences predicted to show above half-maximal fluorescence and the average number of mutations. **d** A low sampling temperature (CoVES T = 0.5, ESM-IF T = 1.0, proteinMPNN T = 1.5) is used while the number of random mutated positions (out of 233 GFP residues) is increased. **e** The maximum number of mutated positions is set to 10, and the sampling temperature is increased. Only combinatorial variants consisting of individual variants on which the surrogate fitness function has been trained on and tested on in panels **a**–**c**, are shown. Source data are provided as a Source Data file.

training (1810 out of 4427 residue mutations at 233 residues due to error-prone PCR library construction limitations), and for the limited range of observed variant mutation numbers (range of mutations: 1–15).

We used CoVES, as well as ESM-IF and proteinMPNN for sampling combinatorial GFP variants at various temperatures and randomly chosen sets of positions to mutate. We then filtered the generated sequences for the ones that the surrogate fitness model can be expected to predict well, and evaluated them for the fraction predicted to be at least half functional, as well as their average mutation distances. Sequences designed with CoVES had a higher fraction of predicted functional sequences and higher average number of mutations than sequences designed by both ESM-IF and proteinMPNN, as well as the measured random error-prone PCR library sequence samples (Fig. 4d, e, Supplementary Fig. 9).

These results indicate that, in the absence of observed combinatorial variant measurements, simple and fast sampling from a per residue, unsupervised model that learns amino acid preferences from the respective amino acid microenvironments could be sufficient to generate functional and diverse combinatorial variants at ten ParD3 antitoxin residues as well as across 233 residues with up to 15 mutations in GFP, and can outperform not just random sampling from experimental libraries but also similar to autoregressive models and their sampling strategies.

## Discussion

We found that a large fraction of observed combinatorial variant effects across 6 proteins, including therapeutically relevant ones, can be explained with an interpretable, simple model that only considers per residue mutational preferences, i.e., without explicit terms for interactions between mutations. While previous studies have described such supervised additive models with a nonlinear transformation (termed 'global epistasis' or 'nonlinear correction' models) for prediction[14–17,26,27] and supervised protein design[21], here we show that this insight can be exploited for generalizable and unsupervised protein design. Specifically, we show that one can effectively design functional and diverse variants just with independent accounting of residue micro-environments using CoVES, and that this approach performs similarly to state-of-the-art high-capacity neural methods in protein design when assessed with surrogate fitness functions. While our assay conditions do not permit measuring gain-of-function antitoxin variants (due to saturation in measured growth rate effects), it is likely that such effects can be captured by such residue preference models. Indeed, the structural microenvironment has been used successfully to identify gain-of-function mutations before[28].

The performance of such mutation preference models, which do not explicitly capture dependencies between mutated residues, does not exclude the existence of higher order epistasis. First, the per residue mutation preferences inherently capture an implicit notion of

dependence on neighboring residues, and we indeed observe that mutations at contacting residues can alter the mutation preferences at a given residue. Second, while 78–98% of the observed combinatorial variant effects can be explained by mutation preferences alone, it is likely that some of the remaining variation in some datasets will be explained by true biological specific dependencies between residues. To identify such dependencies requires a careful treatment of the experimental noise and considerations of statistical power in each selection system, such as done in refs. 11,17, and experimental validation of significant epistatic dependencies, which we leave for future work.

A number of observations suggest that the close structural contexts are the main determinants for variant effect prediction and design: 1) The similarity in performance of CoVES to autoregressive methods that can learn arbitrary mutation dependencies, suggests that the local structural context can capture the majority of predictive effects. 2) We observed that site-wise preferences in the supervised global epistasis models can be altered by mutations in contacting residues. 3) We observed that the explanatory power of the nonlinear, per residue mutation preference model is the least high for combinatorial variant datasets in which spatially close residue are mutated (AAV and 10 position antitoxin dataset). In contrast, the performance is strikingly high for datasets in which strictly or largely non-contacting residues are mutated (i.e. when mutating a small number of random positions spread across an entire domain). 4) Positive deviations from the nonlinear, per residue mutation preference model predictions of combinatorial variants (positive epistasis) have been shown to be in close contact[11], and can indeed be used to fold proteins[29,30]. Together, these observations suggest that future efforts in modeling protein variant effects should focus on considering the structural context of a given protein.

While we demonstrated performance of supervised residue mutation preference models in datasets that contain variants with up to 10, 15, or 21 mutations (for the antitoxin, GFP and AAV, respectively), it will be interesting to examine for which number of mutations and choice of residues such simple residue mutation models can be used. Our observations suggest that choosing residues with non-contacting, 'independent' structural microenvironments will enable simple, per residue mutation effect models to predict exploding numbers of combinatorial variant effects well. Nonetheless, because the supervised mutation effect predictors (surrogate fitness functions) show almost perfect classification accuracy for half-maximal function of GFP and antitoxin variants, we believe that such models can be used as useful additional benchmarks to overcome the limitation of evaluating sequence design methods only by recovery of the wild-type sequence for a given structure[8,25]. We note that, even though such surrogate fitness functions were highly performant in classifying the function of held-out variants, there is the possibility that biases are introduced by using such functions in assessing unobserved sequence variants. Specifically, our particular choice of surrogate fitness function does not consider specific epistatic interactions between mutations, and could therefore be misleading in assessing particular variant combinations in which true biological dependencies are dominant, such as between contacting residues. Additionally, the generality of using this or other choices of surrogate fitness functions for other proteins and residue combinations remains to be determined. Nonetheless, we believe that such surrogate fitness functions are likely to prove useful in choosing among models and sampling temperatures in the absence of expansive experimental evaluation of designed sequences from various models and sampling temperatures.

For the supervised residue mutation effect models, we were able to learn 'rules' for choosing minimum numbers of experimental variant measurements necessary for accurate combinatorial protein variant effect prediction. Specifically, we found that a number of random combinatorial variant observations that five-fold oversample the number of mutational preference parameters (i.e., random observations = N*20*5, where N is the number of mutated residues) was sufficient to give high accuracy in predicting held-out variant effects (>0.80 Pearson correlation coefficient) across 7 combinatorial variant effect datasets, outperforming even the best unsupervised sequence-based variant effect predictors[9]. Choosing a low number of mutated, non-contacting sites could enable predictive tuning of combinatorial variant effects for proteins where high-throughput measurements are not possible, and instead rely on medium throughput methods, such as plate-based screening ones. Even in the case of adeno-associated virus, where neighboring residues are mutated, the supervised non-linear, residue mutation preference model was shown to outperform high-capacity neural models for design[21].

It will be interesting to examine CoVES performance at higher numbers of mutations, or even in full sequence design. In the case of full sequence design, the exact atom-neighborhood around a position is unknown because neighboring residue's side-chains are unknown. Here, it could be possible to use an equivariant graph neural network trained on predicting residue mutation preferences given only the surrounding backbone atoms instead[31]. Alternatively, iteratively choosing residues using CoVES for each site, followed by structural modeling of neighboring side-chain positions could enable generating functional sampled sequences. Additionally, future work could examine using predicted structures[32] instead of experimentally measured structures to infer such per residue mutation preferences.

We note that the assessment of generated sequences depends both on model quality as well as sampling strategies. Improvements in sampling alone could boost the fraction of generated functional sequences. For example, the current sampling algorithm for the ESM-IF model[8] works in an autoregressive manner from the N-terminus without consideration of amino acid identities towards the C-terminus of the currently sampled amino acid.

In sum, our results highlight the power of simple, interpretable residue mutation preference models for explaining observed combinatorial variant effects, their use as a 'surrogate' fitness function to predict unobserved combinatorial variants, even when trained on a small number of observations, and present a structure-based design method, CoVES, for unsupervised protein design applications that can outperform high-capacity approaches.

## Methods

### Bacterial strains, vectors and media
*E. coli* TOP10 strains were grown at 37 °C in M9L medium (1× M9 salts, 100 μM $CaCl_2$, 0.4% glycerol, 0.1% casamino acids, 2 mM $MgSO_4$, 10% v/v LB). Antibiotics were used as follows: 50 μg/ml carbenicillin, 20 μg/ml chloramphenicol in liquid media, and 100 μg/ml carbenicillin, 30 μg/ml in agar plates. The toxins ParE3 was carried as before[11] on the pBAD33 vector (chlor[R] marker, ML3302 for wild-type ParE3) with expression repressed or induced with 1% glucose and L-arabinose at indicated concentrations, respectively, and the antitoxin ParD3 was carried on the pEXT20 vector (carb[R] marker, ML3296) with expression induced by IPTG.

### Antitoxin ten position library construction
To measure the combinatorial variant effects in the antitoxin at ten residues, we constructed two sublibaries: one in which five residues are randomized, and one in which an additional five positions are randomized. In this way, we can guarantee the presence of sufficient number of variants with five or less substitutions, given the curse of dimensionality, in which random sampling at ten positions will generate a distribution of variants for which most will have a high number of substitutions. Both the ten position and five position libraries were constructed using a 2-step overlap-extension PCR protocol[33]. We first used primers DDP704 + DDP142 (see Supplementary Dataset 1) and DDP705 + DDP141 to introduce five randomized positions in the wild-

type antitoxin pEXT20-*parD3* plasmid ML3296 (PCR cycling was: 30 s at 98 °C; 20 cycles of: 10 s at 98 °C, 20 s at 55 °C, 1 min at 72 °C; 2 min at 72 °C, hold at 4 °C; using KAPA). The PCR products were pooled, diluted 1:100 and amplified using the outer primers DDP141 + DDP142 to generate full length, mutated antitoxin sequence. On this PCR product, we then used the primers DDP700 + DDP142 and DDP705 + DDP141 to introduce the next five randomized positions, and used the above strategy to generate full length antitoxin ParD3 gene with 10 positions randomized. We then cloned both the five and ten position randomized PCR product into the pEXT20 vector using restriction digests with SacI-HF and HindIII-HF (NEB) and ligation using T4 DNA ligase (NEB at 16 °C for 16 hours with a 1:3 molar ratio of insert to vector. Ligations were dialyzed on Millipore VSWP 0.025 μm membrane filters for 90 min before electroporating (2 mm cuvettes at 2.4 kV) into TOP10 cells, made using the protocol from ref. 34. Cells were recovered in 1 ml SOC for 1 h. We propagated each library with at least 500,000 transformants, checked by spot plating 1:10 serial dilutions of recovered cells on LB/carb/chlor/1% glucose plates. We grew $OD_{600}$ ~0.5 at 37 °C in M9L/carb/chlor/1% glucose, spun down (8000G, 5 min) and resuspended in 5ml M9L/carb/chlor/1%glucose/20% glycerol for storage at −80 °C. We then made these cells electrocompetent in replicate for each library, and transformed dialyzed wild-type toxin ParE3 into these cells. Cells were propagated with at least 500,000 transformants, and grown up to OD-0.6, before spinning down (8000G, 5 min) and resuspending in 5 ml M9L/carb/chlor/1% glucose/20% glycerol. Cells were aliquoted into 1 ml tubes and flash frozen in liquid nitrogen for storage.

## High-throughput variant effect measurement

On the day of growth rate measurements, aliquots from two separate transformations were thawed and recovered in 50 ml M9L/carb/chlor/1% glucose at 30 °C for 3 h. Subsequently, glucose was removed by washing 4 times with M9L, and cells were ready for growth rate measurement. Growth rate measurements were then performed as described previously[11]. Briefly, washed cells were resuspended in 250 ml M9L/carb/chlor/IPTG to induce antitoxin expression, and toxin expression induced after 100 min by adding arabinose. Cells were diluted 1:10 with pre-warmed media when their $OD_{600}$ reached ~0.3 to keep them in exponential growth throughout the duration of the experiment. 50 ml of the cultures were sampled at the time of toxin induction, and 10 h after. These cells were miniprepped, and a high-input (200 ng plasmid DNA), low cycle (14 rounds) PCR reaction performed to isolate amplicons of interest using primers DDP643-645/DDP648-651/DDP654-657 to introduce Illumina multiplexing indices and adapters. We then gel purified and sequenced these library amplicons as described previously[11]. We then performed sequencing using 250 base-pair paired end reads using a Novaseq SP flowcell for each timepoint and replicate.

## Analysis of high-throughput sequencing data

Paired-end reads were processed as described previously[11]. Briefly, paired-end sequencing reads were merged using FLASH 1.2.11[35]. Merged reads were quality filtered based on their phred-score using vsearch 2.13.0[36], with the following arguments: vsearch --fastq_filter {file_name} --fastq_truncqual 20 --fastq_maxns 3 --fastq_maxee 0.5 --fastq_ascii 33 --fastaout {output_file_name}.fasta. Reads were subsequently filtered for having defined mutations at the desired sites only, and the frequency of each variant at each timepoint was counted. We then calculated a log-read ratio for all variants with at least three reads pre- and post-selection with one pseudocount, and normalized these log-read ratios to fall between 0 and 1 given the log read ratio of truncated and wild-type antitoxin variants. We note that non-functional antitoxin variants that have a growth rate around 0 cannot be resolved. This noise for the non-functional antitoxin variants arises due to sequencing depth limitations in pre- and post-selection

read counts: Variants might drop from different pre-selection read counts to unobserved in the post-selection library and hence show variance in their calculated log read ratios.

## Nonlinear, residue mutation effect modeling of combinatorial variant effects

We used a nonlinear per residue effect model implemented in Tensorflow 2[37] to model combinatorial variant effects in the antitoxin as before[11]. We used one-hot encoding of amino acid variants as a predictor and one additional bias feature, and fit weights associated with each single amino acid mutant substitution as well as the bias parameter. The linear sum of these weights is passed through a sigmoid function to predict the normalized growth rate effect between zero and one for each combinatorial variant in the three and ten position library. We used the Adam optimizer to minimize the mean squared error of predicted to measured normalized log read ratios for each variant, until the training error stabilized. For each protein, a number of learning rates, number of training epochs and batch sizes was tried to minimize the training error. For models fit to GFP, AAV, GRB2, PABP, and GB1, we used an additional modification: The sigmoid output is scaled and shifted with a learnable parameter each to enable modeling experimentally measured values that do not fall within a range between 0 and 1. This also enabled higher flexibility in choosing nonlinear effect. For example, the GFP fitted nonlinearity utilizes only the 'lower' part of the sigmoid output function (Supplementary Fig. 9a). It is possible to find better nonlinear transformations for some of these proteins, but we observed that the scaled and shifted sigmoid function can suffice for high explanatory power (Fig. 2j, Supplementary Fig. 3). We note that this model cannot generalize to combinatorial variants that contain individual substitutions that have not been observed during training, ie. for the GFP dataset, only combinatorial variants that consist of the 1810 observed individual residue mutations (out of a possible 4427 residue mutations at 233 residues) due to error-prone PCR library construction limitations, can be predicted.

## Inference of site-wise preferences from subsampled variant effect measurements and estimation of supervised surrogate fitness function generalization error

To test how well models trained on few observations can predict the remaining held-out observed variants across 7 variant datasets, we retrained the above models on a subset of data and tested their performance on the full dataset. In order to estimate the in-distribution generalization error of predicting unobserved variants, we split the training dataset into 90% training data and assessed the correlation between predicted and observed effects in the remaining 10%, respectively, of held-out test data. We did not consider a validation set since there are no hyperparameters to be optimized. To estimate the generalization error more robustly for the antitoxin 10 position (and 3 position) library, where the majority of combinatorial variants are non-functional, we trained on an 80%-20% train test split, to increase the number of test examples that are able to neutralize the toxin. Similarly, to estimate the error in classifying variants that achieve at least half-maximal function in the antitoxin or GFP, we calculated area under the receiver operator curves and positive predictive values (true positives/(true positive + false positives)) on the random held-out test sets.

## Inferring changes in antitoxin ParD3 site-wise mutation preferences by contacting residues

A previous study has examined the effect of the combinatorial three position antitoxin binding residue library (at antitoxin residues D61, K64 and E80) in the presence of various toxin variants (single residue toxin variants N99V, V75C, D55E, A66I, V5L, A66F, R100W, E87M)[11]. To examine the site-wise preferences at each antitoxin position in each

toxin variant background, we fitted separate site-wise logistic regression models to the antitoxin library in each toxin variant background. The inferred antitoxin residue mutation preferences were almost perfectly correlated across 9 out of 10 toxin backgrounds, in which toxin substitutions do not contact any of the mutated antitoxin residues, but deviated in the background of toxin E87M (Supplementary Fig. 5). Inspection of the site-wise preferences revealed deviations specifically for antitoxin residue K63, which is found to be directly contacting the toxin E87M position (Supplementary Fig. 5a).

### Estimating the total number of functional variants

We estimated the total number of functional antitoxin sequences with at least half-maximal neutralization of the toxin using two orthogonal approaches, with both approaches predicting ~$10^{10}$ functional sequences. The first estimate is based on the empirically measured distribution of fitness effects (Supplementary Fig. 1). Here, we calculated the fraction of sampled sequences that are measured to be functional at each mutation distance from the wild-type sequence, and multiplied this fraction by the total number of sequences that exist at this mutation distance. The total number of functional variants is the sum of all variants at each mutation distance. This approach assumes that the sampled sequences are a representative sample of all possible random mutations at each mutation distance, and does not consider the noise in the estimate of the fraction of functional sequences. For example, there are no functional sequences among the set of variants that show seven substitutions, likely due to the lower number of variants sampled at this mutation distance.

To address these issues, we also used the supervised, ten position surrogate fitness function to estimate the total number of functional sequences. Here, we generated one million random synthetic combinatorial variants, and asked the surrogate fitness function to predict what fraction of these variants reaches half-maximal growth rate, p(predicted growth rate (GR) > 0.5). We then estimated the fraction of true positive predictions, p(observed GR > 0.5|predicted GR > 0.5), by sampling with replacement the observed variants, for which observed and predicted growth rate data are available, to match the distribution of predicted growth rates for the synthetic sequences. Then we calculated the fraction of these subsampled observed variants with GR > 0.5 among variants with predicted GR > 0.5. Finally, we multiplied these two fractions (p(predicted GR > 0.5) and p(observed GR > 0.5| predicted GR > 0.5)) to estimate the fraction of synthetic variants that are both predicted and observed to be functional, p(observed GR < 0.5, predicted GR > 0.5). We then multiplied this probability by the total number of possible variants ($20^{10}$) to estimate a lower bound on the total number of functional sequences.

### Unsupervised protein variant scoring

To implement CoVES, we trained the RES classifier model as described previously[22]. Briefly, this equivariant graph neural network is trained on predicting the amino acid identity of a masked residue given ~500 surrounding atoms, excluding hydrogens, from the protein database (PDB). Importantly, the training and test set were split according to domain-level structural CATH topology[23,25], and training examples were down-sampled to the least common amino acid to prevent biased learning of the classifier. The trained model achieved ~53% accuracy on the held-out test set.

To score combinatorial variants for the antitoxin, we fed the wild-type structural environments, including both toxin and antitoxin atoms in their biological octameric assembly, surrounding each mutated antitoxin residue to the graph neural network and obtained classifier scores for each amino acid at each positions. We then summed the log probability normalized classifier scores at each site (log $p(amino\ acid)\alpha\frac{-amino\ acid\ score}{t}$) using t = 0.1) to result in a combinatorial variant effect score. We followed an analogous approach for GFP (PDB ID: 2WUR), GRB2 (PDB ID: 1GCQ) and AAV (PDB ID: 1LP3). To

score mutants in GFP, GRB2 and AAV using proteinMPNN and ESM-IF, we calculated the conditional autoregressive log-probabilities for each of the sequences further conditioned on the PDB structure.

### Unsupervised protein variant generation

To sample antitoxin variants using CoVES, we converted the structure-learned amino acid preference scores into probabilities at each site using the Boltzmann distribution ($p(amino\ acid)\alpha e^{\frac{-amino\ acid\ score}{t}}$). Using equal weighting between sites, we sampled 500 random sequences at varying temperatures (t ∈ {0.1, 0.5,0.7,1,1.5,2,2.25,2.5,2.75,3,4,5}) at each mutated site and deduplicated the sampled sequences for evaluation. Model weights and sampling strategies for proteinMPNN and ESM-IF followed previous studies[8,24]. For proteinMPNN, we generated 30 sequences at each temperature (t ∈ {0.1, 0.3, 0.5, 0.7, 0.8, 0.9, 1, 1.1, 1.2, 1.3, 1.4, 1.5}). For ESM-IF, we generated 100 sequences and deduplicated these variants at each temperature (t ∈ {0.1, 0.3, 0.5, 0.7, 0.8, 0.9, 1, 1.1, 1.2, 1.3, 1.4, 1.5}). For all models, we fixed amino acids that were not mutated, and also tied any mutations in corresponding positions between the four antitoxin chains in the octameric toxin-antitoxin sequence from PDB ID: 5CEG. We then evaluated the growth rate effects of generated variants using the supervised surrogate fitness function trained on the ten position library data, which is estimated to have low generalization error when predicting unobserved variant effects (Fig. 2). We used EvCouplings to score antitoxin variants as done previously[11]. We then sampled sequences, while fixing all non-mutated positions, by performing simulated annealing with a temperature cycle of 1.0*scaling_factor, 0.5*scaling_factor, and 0.2*scaling factor, each for 160 steps. We note that EvCouplings is not able to generate samples for the antitoxin position L48, as this column does not have sufficient coverage in the alignment to infer robust site-wise and coupling parameters. In order to compare these sampled sequences fairly between models, we subsampled the number of unique generated variants from all models to 30, which is the number of unique sequences sampled across most temperatures for the proteinMPNN model. Finally, to help in visualization of Fig. 3, we fit polynomial fits of various polynomial order to each model's generated sequence summary statistics.

### Experimental evaluation of CoVES designed sequences

To validate sampled sequences from CoVES, we generated 12 random sequences with mutations at the above defined 10 antitoxin positions from the set of unique sequences sampled at t = 1.5. Additionally, we also asked whether CoVES is able to design functional variants at each hamming distance from 2, 3, 4, 5, 6, 7, 8, 9 and 10 from wild-type. In order to do so, we also chose the CoVES designed variants that had the highest surrogate fitness function score at each mutation distance. After deduplication, this led to a final total set of 17 antitoxin variants, with antitoxin versions 1–12 comprising the randomly sampled sequences (see Supplementary Dataset 2), and antitoxin versions 15, 16, 17,18, 19 comprising surrogate-fitness function biased variants. These were ordered from IDT as ultramers (see Supplementary Dataset 2), individually amplified using PCR DDP330 and DDP331 using KAPA HiFi master mix (98 °C for 3 min; 31 cycles of: 20 s at 98 °C, 15 s at 65 °C, 15 s at 72 °C; 1 min final extension at 72 °C, and 4 °C hold). The antitoxin pEXT20-parD3 plasmid ML3296 was used to amplify a backbone vector using DDP332 and 333 (98 °C for 3 min; 31 cycles of: 20 s at 98 °C, 15 s at 65 °C, 4 min at 72 °C; 5 min final extension at 72 °C, and 4 °C hold)., and subjected to DpnI digestion (NEB, 1 h at 37 °C). The amplified ultramer was then introduced into the backbone vector using the NEBuilder HiFi DNA Assembly Master Mix according to manufacturer protocol. This Gibson reaction was then transformed into Top10 cells, selected on LB/carb plates, and miniprepped plasmid from individual colonies was sequenced using DDP141 to verify the correct mutation. For one designed antitoxin variant (antitoxin version 4), an incorrect mutation was found, so it was left out in the

subsequent steps. Finally, mutated antitoxin variants were each co-electroporated with wild-type toxin containing plasmid from ML3302 into Top10 cells and selected on LB/carb/chlor/1% glucose plates. Individual colonies were grown overnight at 37 °C in LB/carb/chlor/1% glucose liquid media, and then serially diluted 1:10, before spotting on LB/carb/chlor/1%glucose plates (repressive conditions), LB/carb/chlor/0.2% arabinose (toxin-inducing conditions) and LB/carb/chlor/0.2% arabinose/100uM IPTG (toxin and antitoxin expressing conditions). Plates were grown overnight at 37 °C before recording their final titers.

### Unsupervised GFP variant generation and evaluation
GFP variants were generated using CoVES, proteinMPNN and ESM-IF by both varying the sampling temperature, as well as the number of randomly chosen residues (among the following mutation ranges: {5, 6, 7, 8, 9, 10, 11, 12, 13, 14, 15, 16, 17, 18, 19, 20}) that are designed, with the remaining positions fixed to their wild-type amino acid. The examined CoVES sampling temperatures were among {0.1, 0.3,0.4, 0.5, 0.6, 0.7,1,2, 3, 4, 5, 8, 10, 20}. The examined ProteinMPNN sampling temperatures were among {0.0001, 0.1, 0.15, 0.2, 0.25, 0.3, 0.5, 0.7, 1.0,1.5,1.75, 2,2.25, 2.5,5,10}, and ESM-IF sampling temperatures among {1e-2,1e-1,0.5,0.75,1, 1.25, 1.5, 1.75, 2, 3, 4, 5, 10}. In order to ensure that only designed variants are assessed which are not out of distribution and fall within the set of sequences that the surrogate fitness function has predictive power on and has been evaluated on during random 90%-10% testing, designed variants were then post-filtered to a) contain fewer than 15 residue mutations and b) contain individual variants that have been observed by the surrogate fitness function when trained on the observed combinatorial variant dataset (1810 per residue variant effects among 233 residues).

### Supervised protein variant generation
To report on the number of functional variants at various mutation distances from the random library (Fig. 3e), we fetched all the observed library observations, and calculated the function of observed variant effects at a given hamming distance from the wild-type sequence, without sampling.

To sample from the supervised, logistic regression 'surrogate' fitness function, we converted site-wise preferences fit to the total combinatorial dataset into energies by min-max normalization and using the Boltzmann distribution to obtain probabilities for each amino acid variant at each position. We then used various temperatures (t ∈ {0.5, 0.7, 0.8, 0.9, 1, 1.5, 2}) to generate sets of sampled sequences.

### Statistics & reproducibility
No data were excluded from the analyses. No statistical method was used to predetermine sample size. The Investigators were not blinded to allocation during experiments and outcome assessment.

### Reporting summary
Further information on research design is available in the Nature Portfolio Reporting Summary linked to this article.

### Data availability
All relevant data supporting the key findings of this study are available within the article and its Supplementary Information files. The raw variant effect measurement data generated in this study have been deposited in the sequencing read archive database under accession code PRJNA988522. The processed variant effect measurement data are available at github[38] as well as in the Source Data file.

Previously published deep mutational scanning data sources are from:
- antitoxin 3-position library[11]:
- GB1[18]:
- PABP:[19].

- GRB2[20]:
- AAV[21]:
- GFP_SAR[16]:
- GFP_POE[17]:
Used PDB IDs: 5CEG, 1LP3, 1PGA, 1FCC, 2WUR, 2VWF, 1CVJ. Source data are provided with this paper.

### Code availability
Software is available at: Ding, David & Shaw, Ada. Protein design using structure-based residue preferences, CoVES, https://doi.org/10.5281/zenodo.10461017, 2024.

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

## Acknowledgements

We thank members of the Laub, Marks and Savage labs for helpful feedback, and Chloe Hsu for helpful discussions on sampling from ESM-IF. We thank Joseph Rivera with help screening colonies of designed antitoxin variants. We thank Jennifer Listgarten for sug-gesting the term 'surrogate fitness function'. This work was supported by the Howard Hughes Medical Institute (MTL, DFS), National Insti-tutes of Health grant R01CA260415 (DSM) and Chan Zuckerberg Initiative CZI2018-191853 (DSM).

## Author contributions

Conceptualization, Methodology, Validation, Formal analysis, Investi-gation, Writing – original draft and preparation: D.D. Data Curation, Resources: D.D., A.Y.S. Software: D.D., A.Y.S, N.R. Writing – review and editing: D.D., S.S., D.S.M., M.T.L., D.F.S., N.P., A.Y.S. Supervision, Project Administration: D.S.M., M.T.L., D.F.S. Funding Acquisition: D.S.M., M.T.L., D.F.S.

## Competing interests

SS is employed by Dyno Therapeutics. DSM is an advisor for Dyno Therapeutics, Octant, Jura Bio, Tectonic Therapeutics, and Genentech, and a co-founder of Seismic. NR is employed by Seismic. DFS is a co-founder and scientific advisory board member of Scribe Therapeutics. The remaining authors declare no competing interests.
