## [Peer Review File · Nature Communications]

Protein design using structure-based residue preferencesReviewer #1 (Remarks to the Author):

Ding et al present a highly interesting study that investigates the capacity of per residue (position-specific) amino acid preferences to predict the effect of combinatorial mutations. The results are remarkable in that the relatively simple per-site potentials (one supervised, one unsupervised and structure based) often out-perform or perform on par with vastly more complicated neural network based approaches. Together the data provide an important benchmark establishing the performance of site-specific potentials to explain high-order data, and motivate future work to explore simpler, more interpretative models describing protein sequence-function relationships.

Major comments:

While I really enjoyed the manuscript, at times I felt the results were somewhat overstated or presented in an over-general way. I would appreciate a more nuanced discussion of when CoVES is expected to well-capture the data and when it does not. More specifically:

- The model was assessed on 7 different datasets, however many of these are focused on protein binding/complex formation. Is the model expected to do as well for enzymes or allosteric systems where long-range cooperative interactions are thought to be important?
- Following on this, it would be useful to include more information on the datasets themselves in the main text (maybe as a table). This is necessary to provide context for interpreting the authors results. For example, if the authors are predicting the effect of five non-contacting mutations dispersed throughout the structure, the finding that an interaction-free potential works well is not surprising. Relevant information on the dataset includes: order of the mutations (how many were made in combination?), localization of mutations (average distance of mutated residues, largest number of mutations that physically contact one another, proximity to the binding or functional site), and perhaps some sense of the noise in each dataset.
- The performance isn't equal across all datasets; the AAV dataset is less well-predicted by the supervised model (figure 2j) the gb1 and AAV datasets seem to require more training data for the supervised model (figure 2k), and AAV and GRB2 are harder for CoVES (and many of the other undersupervised methods) to capture. In the discussion, the authors touch on the idea that the AAV data are harder to predict because spatially close residues are mutated. This seems like an important point to develop further in the results. Again, if the test sets were biased towards non contacting residues distributed throughout the structure, the authors results become less remarkable.
- The authors point out that "choosing special residues will effect performance" and provide as an example 13 active site residues of GFP with known higher order interactions. Have the authors tried to predict these data and verified that the performance was poor? It would be useful to consider/show a case where the simple site specific models fail. This would give more nuance to the paper.
- The abstract states that "this simple biologically rooted model can be an effective alternative to high-capacity, out of domain models for the design of functional proteins". While the data do indicate that CoVES can identify or design functional mutant combinations, the correlations are sometimes poor, and the task of designing mutant combinations for a subset of residues is less ambitious than total sequence design. It may be better to rephrase to "this simple biologically rooted model can be an effective alternative ... for the identification of functional combinatorial mutants"
- In the discussion the authors write that "the performance of CoVES, which samples and scores residue preferences solely on the local structural context, over autoregressive methods... suggests that the structural context is sufficient". However, the spearman correlation coefficients presented for CoVES in figure 3b are fairly modest (particularly for AAV and GRB2). To me, this suggests that local structural context does as well or better than autoregressive methods, but that none of the unsupervised methods are (yet) completely sufficient for high accuracy prediction and/or design.

Minor comments:

- I am confused about the details of the toxin-antitoxin assay in figure 2. The equation between panels B and C seems to suggest that growth rate was calculated as proportional to the relative enrichment at 10 hours (at a single time point), but the graph of read count over time (blue lines just below) and the figure legend (which mentions "high throughput sequencing over time") seem

to indicate that multiple time points were taken. Please clarify the growth rate calculations in figure 2 – are they based on one time point, or fitting across many?

Reviewer #2 (Remarks to the Author):

Ding et al. present data consistent with the premise from prior literature that epistasis is present among mutations but sparse. They show in low-density combinatorial datasets that knowledge on the first order effects of mutations is sufficient to capture much but not all of the variation in full- or high-order combinatorial spaces. They then develop an unsupervised structure-based algorithm to predict these first-order effects of mutations and propose that this algorithm can efficiently predict functional combinatorial variants. The critical problem in this manuscript is that there is no experimental validation of any of the generated variants, from what I can tell. Instead, it appears that all of the benchmarking is against predicted functionality of variants from a main-effects / non-epistatic model, and they conclude that a main-effects / non-epistatic model is therefore a better performer than other unsupervised algorithms. This seems to be a fundamental flaw that hinders understanding of the performance of the new unsupervised model and its performance versus other algorithms that do consider sequence context, which must be tested against “real” data where context-dependency is allowed.

More detailed comments:

To what extent are these results dependent on limitations in the experimental datasets e.g. in how densely or sparsely they focus on residues enriched for their epistatic interactions. It is notable that the main-effects models perform significantly better on the 3-residue set than the 10-residue set – suggests that as the scale grows to “all sites”, the increased chances of epistasis make the problem more difficult, and in converse that the impressive performance on the 3-residue dataset is not generalizable? (Indeed, even in these two datasets the generalizability of the quality predictions in the 3-residue set do not extend to the 10-residue set, yet the authors later on focus only on the “near-perfect” performance of the former and sweep the poorer performance on an expanded dataset under the rug). Extending this train of thought, the commonly-used-benchmark double mutant GB1 DMS dataset shows that most pairs of mutations don't interact epistatically, but virtually all mutations have at least one other mutation out there with which they do interact epistatically. Therefore, a main-effects only model will fail in certain contexts for every mutation, and you just don't know when. Overall, I think the use of “oracle” to describe this main-effects model, when it achieves an R2 of only 0.81 for one of the two primary datasets, is a bit of an overstatement. The authors do briefly discuss related ideas to this critique in the paragraph starting at line 332, but that doesn't eliminate this as a shortcoming of the work.

In sections 3c-e, the use of one set of predicted scores (which are based on a model that had R2 of 0.81 with the actual measured mutants from within the space) to benchmark another set of predictions (which had R2 of ~0.4 with the actual measured mutants from within the space) seems flawed -- especially because the proposed model and its benchmark evaluator share an error mode in that neither will account for the sequences in this landscape that truly do have epistasis. The weirdness of this setup is amplified when you benchmark methods that do incorporate sequence context/epistasis against this same dataset generated without epistatic terms – it is perhaps not surprising that COVES “outperforms” these other models because it is going to better match the error mode of the ‘benchmark’ evaluator. This benchmarking really needs to be done with experimentally validated sequences. Even though the beginning of the GFP section in Figure 4 is premised on the idea that there are these higher-number combinatorial variants in the library, the main-effects LogReg model is again used as the benchmarking standard for evaluating function of generated sequences which seems problematic.

Details on library characteristics are sparse. Is this complete combinatorial complexity in the sampled positions, or low-order-biased mutational numbers across the possible 10 positions? What is the average and distribution of number of mutations per variant?

Response to Reviewers for Ding et al.: Protein design using structure-based residue preferences

Reviewer #1 (Remarks to the Author):

Ding et al present a highly interesting study that investigates the capacity of per residue (position-specific) amino acid preferences to predict the effect of combinatorial mutations. The results are remarkable in that the relatively simple per-site potentials (one supervised, one unsupervised and structure based) often out-perform or perform on par with vastly more complicated neural network based approaches. Together the data provide an important benchmark establishing the performance of site-specific potentials to explain high-order data, and motivate future work to explore simpler, more interpretative models describing protein sequence-function relationships.

Major comments:

While I really enjoyed the manuscript, at times I felt the results were somewhat overstated or presented in an over-general way. I would appreciate a more nuanced discussion of when CoVES is expected to well-capture the data and when it does not. More specifically:

- The model was assessed on 7 different datasets, however many of these are focused on protein binding/complex formation. Is the model expected to do as well for enzymes or allosteric systems where long-range cooperative interactions are thought to be important?

How well the performance of such residue-wise models generalizes to other classes of proteins is a highly interesting question. However, we are not aware of combinatorial variant datasets in enzymes or allosteric systems that would enable testing this hypothesis in a way that allows for separating other confounding issues, such as differences in experimental reproducibility between datasets which could cause differences in model performance (see below). So, while we believe that such residue-wise approaches should be generally applicable, we leave the experimental demonstration of this for future work.

- Following on this, it would be useful to include more information on the datasets themselves in the main text (maybe as a table). This is necessary to provide context for interpreting the authors results. For example, if the authors are predicting the effect of five non-contacting mutations dispersed throughout the structure, the finding that an interaction-free potential works well is not surprising. Relevant information on the dataset includes: order of the mutations (how many were made in combination?), localization of mutations (average distance of mutated residues, largest number of mutations that physically contact one another, proximity to the binding or functional site), and perhaps some sense of the noise in each dataset.

Thank you for your comments. We have now updated the Supplementary Table 2 to include this information and moved this information into the main manuscript.

- The performance isn't equal across all datasets; the AAV dataset is less well-predicted by the supervised model (figure 2j) the gb1 and AAV datasets seem to require more training data for the supervised model (figure 2k), and AAV and GRB2 are harder for CoVES (and many of the other undersupervised methods) to capture. In the discussion, the authors touch on the idea that the AAV data are harder to predict because spatially close residues are mutated. This seems like an important point to develop further in the results.

We find the causes of different model performance across datasets highly intriguing as such reasons could enable predicting model performance in new combinatorial variant datasets. However, differences in performance across datasets could be due to a variety of factors, such as differences in real biological epistasis, differences in experimental measurement noise between datasets, the particular set and distribution of measured mutations, as well as other factors. Because we cannot remove confounding factors for predicting model performance, we prefer to speculate on these factors in the Discussion section, rather than in the Results section.

Again, if the test sets were biased towards non contacting residues distributed throughout the structure, the authors results become less remarkable.

Our test sets were a random set of all the combinatorial variants, so are not expected to have any bias towards non-contacting residues compared to the training set in each dataset.

- The authors point out that “choosing special residues will effect performance” and provide as an example 13 active site residues of GFP with known higher order interactions. Have the authors tried to predict these data and verified that the performance was poor? It would be useful to consider/show a case where the simple site specific models fail. This would give more nuance to the paper.

Thank you for suggesting this. We have now performed this analysis (updated Fig. S3, Table 1 and Fig 2j with Dataset “GFP_POE”), and find that even when 13 active site residues are mutated, the nonlinear, residue-wise preference model alone can explain 94% of all the observed mutant variants (ie. using ~27 parameters to explain ~8000 combinatorial variants in this dataset).

- The abstract states that “this simple biologically rooted model can be an effective alternative to high-capacity, out of domain models for the design of functional proteins”. While the data do indicate that CoVES can identify or design functional mutant combinations, the correlations are sometimes poor, and the task of designing mutant combinations for a subset of residues is less ambitious than total sequence design. It may be better to rephrase to “this simple biologically rooted model can be an effective alternative ... for the identification of functional combinatorial mutants”

Thanks, we changed this.

- In the discussion the authors write that “the performance of CoVES, which samples and scores residue preferences solely on the local structural context, over autoregressive methods... suggests that the structural context is sufficient”. However, the spearman correlation coefficients presented for CoVES in figure 3b are fairly modest (particularly for AAV and GRB2). To me, this

suggests that local structural context does as well or better than autoregressive methods, but that none of the unsupervised methods are (yet) completely sufficient for high accuracy prediction and/or design.

Thank you for pointing this out. We have changed this sentence to be more moderated.

Minor comments:

- I am confused about the details of the toxin-antitoxin assay in figure 2. The equation between panels B and C seems to suggest that growth rate was calculated as proportional to the relative enrichment at 10 hours (at a single time point), but the graph of read count over time (blue lines just below) and the figure legend (which mentions “high throughput sequencing over time”) seem to indicate that multiple time points were taken. Please clarify the growth rate calculations in figure 2 – are they based on one time point, or fitting across many?

We apologize for the confusing wording. We have now clarified the legend to reflect that exactly 2 timepoints were taken. We prefer to keep the graph of read counts over time (blue lines in Fig. 2b) for illustrative purposes.

Reviewer #2 (Remarks to the Author):

Ding et al. present data consistent with the premise from prior literature that epistasis is present among mutations but sparse. They show in low-density combinatorial datasets that knowledge on the first order effects of mutations is sufficient to capture much but not all of the variation in full- or high-order combinatorial spaces. They then develop an unsupervised structure-based algorithm to predict these first-order effects of mutations and propose that this algorithm can efficiently predict functional combinatorial variants. The critical problem in this manuscript is that there is no experimental validation of any of the generated variants, from what I can tell. Instead, it appears that all of the benchmarking is against predicted functionality of variants from a main-effects / non-epistatic model, and they conclude that a main-effects / non-epistatic model is therefore a better performer than other unsupervised algorithms. This seems to be a fundamental flaw that hinders understanding of the performance of the new unsupervised model and its performance versus other algorithms that do consider sequence context, which must be tested against “real” data where context-dependency is allowed.

We would like to thank the reviewer for suggesting experimental validation of designed sequences as the true benchmark of sequence design models. In this vein, we have now experimentally tested 16 antitoxin variants designed using CoVES and validated that 10 of these are indeed functional (see edits on page 14 and Fig. 3c). These 10 functional antitoxin variants are diverse, having up to 9 mutations with respect to the wild-type antitoxin. These results strengthen our conclusion that CoVES can indeed design diverse and functional protein variants.

Please see our detailed comments below addressing concerns regarding the performance assessment of CoVES vs. other unsupervised methods. In brief, we agree with the reviewer's comments that comments about the performance of CoVES vs. other methods could be sensitive to errors made by the surrogate fitness function (even though minimal for half-maximal classification, see below). We have hence moderated statements about the performance of CoVES compared to other sequence design methods (stating for example that ‘CoVES performs similarly to other methods when evaluated with the surrogate fitness function’, rather than ‘CoVES outperforms other methods.’, see edits on page 2,4,18).

However, we do believe that using surrogate fitness functions, like the supervised functions we employed in this study, are a useful approach to suggest and hint at performance differences between sequence design models. We chose to be careful in using the surrogate fitness functions not for continuous prediction, but the much more robust task of classifying half-maximal function, for which our surrogate fitness functions achieve very high classification accuracy on the test set of held-out variants (antitoxin AUROC: 0.97, GFP AUROC: 0.99). Unfortunately, the true experimental validation of >10.000s of designed sequences from various models and temperatures is an unprecedented effort that is out of scope of this work. In light of this, we believe that our approach is one principled form of choosing among sequence design methods to address a challenge faced by many practitioners of protein design.

More detailed comments:

To what extent are these results dependent on limitations in the experimental datasets e.g. in how densely or sparsely they focus on residues enriched for their epistatic interactions. It is

notable that the main-effects models perform significantly better on the 3-residue set than the 10-residue set – suggests that as the scale grows to “all sites”, the increased chances of epistasis make the problem more difficult, and in converse that the impressive performance on the 3-residue dataset is not generalizable?

This is a highly intriguing question. Differences in performance between datasets could arise from a number of factors such as differences in reproducibility between the experiment, the number of mutated residues as the reviewer suggests, the proximity of the mutated residues, etc.

When considering all 8 datasets, we notice that the 10 position antitoxin and AAV dataset are least well explained ($R^2 \sim 80\%$) by a main-effects model. These are the 2 datasets that differ from the others in that many neighboring positions are combinatorially mutated. In contrast, all other datasets in which either strictly (3 defined position in the antitoxin), or mostly non-contacting (small number of random mutations across an entire protein, which tend not to contact by chance) residues are mutated, show higher performance ($R^2 \sim 90\%$ or higher). This suggests that performance is mainly driven by whether mutations are contacting or not, rather than the absolute number of mutations introduced per se, and, because we observe this explanatory power across multiple proteins, suggests that this approach should generalize to other sets of non-contacting residues in different proteins. However, we are not able to rule out the contribution of other factors, such as differences in experimental measurement noise, and prefer to leave this speculation in the Discussion section as is.

(Indeed, even in these two datasets the generalizability of the quality predictions in the 3-residue set do not extend to the 10-residue set, yet the authors later on focus only on the “near-perfect” performance of the former and sweep the poorer performance on an expanded dataset under the rug).

We apologize for the confusing wording. With ‘near-perfect’ oracle function, we refer to using the surrogate fitness function trained on the observed data for classifying half-maximal antitoxin function on the held-out set (AUROC = 0.97, Fig. 2i bottom), rather than the continuous prediction. We chose this classification task for evaluating designed sequences, because of its increased robustness to errors in the continuous predictions by the surrogate fitness function. We have now changed the wording to explicitly highlight that the oracle function is used for classification of half-maximal function of designed sequences on page 14.

Extending this train of thought, the commonly-used-benchmark double mutant GB1 DMS dataset shows that most pairs of mutations don’t interact epistatically, but virtually all mutations have at least one other mutation out there with which they do interact epistatically. Therefore, a main-effects only model will fail in certain contexts for every mutation, and you just don’t know when.

We cannot, and do not claim that all combinatorial variant effects are explained by the main-effects model. However, we believe that explaining 95% of the observed $\sim 536,962$ combinatorial variant effects with just the per residue mutation effects ($\sim 1,158$ parameters) in GB1 is a noteworthy result, even if there are occasional, significant deviations from such predictions. In order to capture the remaining 5% of observed combinatorial mutant effects that potentially do depend on significant epistatic effects, one would have to estimate the remaining

530,400 pairwise terms - a highly data-inefficient task. Finally, the high explanatory power of such models across datasets has enabled us to develop rules for how few observations are needed to achieve good predictive performance as well as an unsupervised, structure-based sequence design method that can generate functional and diverse sequences.

Overall, I think the use of “oracle” to describe this main-effects model, when it achieves an R2 of only 0.81 for one of the two primary datasets, is a bit of an overstatement.

Thank you. In order to prevent confusion, we have changed the word ‘oracle’ to ‘surrogate’ fitness function now.

The authors do briefly discuss related ideas to this critique in the paragraph starting at line 332, but that doesn’t eliminate this as a shortcoming of the work.

In sections 3c-e, the use of one set of predicted scores (which are based on a model that had R2 of 0.81 with the actual measured mutants from within the space) to benchmark another set of predictions (which had R2 of ~0.4 with the actual measured mutants from within the space) seems flawed -- especially because the proposed model and its benchmark evaluator share an error mode in that neither will account for the sequences in this landscape that truly do have epistasis.

Thank you for bringing up this concern. Since we only use the supervised function (with R2 of 0.81 for observed variants) of the 10 position antitoxin variant library in a classification task of half-maximal fitness when assessing designed sequences, we believe that the correct evaluation metric relates to how well this model is able to classify variants that do or do not achieve half-maximal fitness. For both sequence design evaluations in antitoxin and GFP, the AUROC is extremely high: 0.97 or 0.99, respectively. Because this classification accuracy is so high, we expect that using this surrogate fitness classifier is the best we can do in the absence of further extensive experimental efforts (see below).

The weirdness of this setup is amplified when you benchmark methods that do incorporate sequence context/epistasis against this same dataset generated without epistatic terms – it is perhaps not surprising that CoVES “outperforms” these other models because it is going to better match the error mode of the ‘benchmark’ evaluator. This benchmarking really needs to be done with experimentally validated sequences.

Thank you for raising this concern. As mentioned above, we have now experimentally validated CoVES for being able to generate functional antitoxin variants.

We agree with the reviewer’s comments that the true comparison of methods would rely on experimental validation of designed sequences for each model and each temperature (since sampling temperatures do not have a 1:1 correspondence between models), and subsequently performing the entire deep mutational scanning experiment with these 10,000s of new variants. This is an unprecedented and costly effort that is out of scope for this work.

In the absence of this, we believe that using highly performant (AUROC ~ 0.98) surrogate functions can be a helpful tool for deciding on sampling models and temperatures. For example, our ability to generate diverse and functional antitoxin sequences (updated Fig. 3c) highlight the usefulness of using such a surrogate function to decide on a sampling temperature (ie. choosing sampling $t=1.5$ for CoVES) to generate functional and diverse antitoxin sequences (updated Fig.

3c). Additionally, the surrogate fitness scores are well correlated with the observed ability of designed antitoxin variants to neutralize the toxin (Fig. 3c right column).

We appreciate that even though the performance of our surrogate fitness function for classification is high, there is, like with any non-perfect classifier, possibility for error and bias. To mitigate the misinterpretation of our results, we have now reworded the concluding performance evaluation of the methods with the surrogate functions to be more suggestive/tentative (ie. instead of 'clearly outperforming other methods' to be 'suggestive of/expected to be performing highly/generating putatively functional sequences' of CoVES (ie. page 15))

Even though the beginning of the GFP section in Figure 4 is premised on the idea that there are these higher-number combinatorial variants in the library, the main-effects LogReg model is again used as the benchmarking standard for evaluating function of generated sequences which seems problematic.

Thank you for raising concerns regarding the main-effects logistic regression model applied to the GFP dataset. We note that we only use this surrogate fitness function trained on the data to classify half-maximal function of designed sequences, and that the estimated generalization error for this classification task on random held-out variants with higher-number combinatorial variants (up to 15 mutations) is very low (AUC: 0.99, Fig. 4). We therefore believe that this surrogate function is appropriate to suggest which model and sampling temperature is better at designing diverse and functional variants. As above, like any non-perfect classifier, there is possibility for error and bias, but we believe that the utility of this cheap and available assessment is the best thing we are able to do in the absence of extensive experimental efforts. To mitigate misinterpretation of our results, we have changed the description of these results to be more tentative/suggestive.

Details on library characteristics are sparse. Is this complete combinatorial complexity in the sampled positions, or low-order-biased mutational numbers across the possible 10 positions? What is the average and distribution of number of mutations per variant?

These details can be found in the Methods section (page 25), Table 1 (formerly Supplementary Table 2) and Supplementary Figure 1.

Reviewer #1 (Remarks to the Author):

Thank you for an effective revision. I appreciated the inclusion of new experimental data (the designed antitoxins). I also was surprised and by the interesting result that CoVES works so well on the GFP_POE data set (which includes 13 active site residues, some of which display higher order epistasis). I agree with the idea that a surrogate fitness function (I like this wording much better than oracle, btw) is a useful strategy to compare sampling models at different temperatures in the absence of expansive (and expensive) experiments. This suggests an interesting strategy for other authors in future work, and I like the idea that this could be used to focus experiments around models and temperatures that are most likely to be successful or informative.

As a very minor comment, it appears that "surrogate" was swapped out for "oracle" with a search and replace; as a consequence in some spots the articles don't match up ("an surrogate fitness function").

My only remaining request regards the code and software submission checklist, which seems incompletely satisfied.

It is great that the authors offer data and source code on git hub. The github repo seems reasonably well organized, however after poking around a little, my sense is that I would have a hard time getting started with it myself despite being a reasonably seasoned coder. More specifically:

- I don't see a clear demo or use instructions – the README table of contents states that there is code to run on the manuscript data BUT not where it is or what to run. It would be good to add a top level file that outlines the general workflow for how to run the code on the data to obtain results in the paper or otherwise demo the code. To put it another way, if I wanted to try out the code on a new protein or data set, where would I start, and what order would I need to call the scripts in? Something as simple as knowing the order to do things in and paths to find the relevant files would be a big help.

- I also don't see the requested information on "expected output" or "expected runtime" as specified on the software submission checklist. It would be good to know what outputs should appear in what directories if running the code is successful. While I fully understand that specifying general runtimes is near impossible (as it depends on machine architecture/environment), it would be useful to say approximately how much time the code took to run in your hands given a particular compute environment (specifying number of processors/memory requirements/GPUs vs CPUs would help here).

To summarize: the github repo is already quite good and doesn't need a major overhaul, but providing a little bit more documentation would go a long way in increasing accessibility/usability

Reviewer #2 (Remarks to the Author):

The comments in my original review were largely addressed.

'oracle' still used in Figure 1, Figure 4 text

Some figure subpanel references for Fig. 3 need to be updated with the new Fig. 3c insertion

I still think the issue of potential bias in the surrogate fitness function being used for comparison of different design algorithms/tools is an important point that the authors responded to in the response letter but didn't add to any discussion or caveat section of the manuscript that I see. This still seems to me like something that should be more explicitly discussed in the text.

REVIEWER COMMENTS

Reviewer #1 (Remarks to the Author):

Thank you for an effective revision. I appreciated the inclusion of new experimental data (the designed antitoxins). I also was surprised and by the interesting result that CoVES works so well on the GFP_POE data set (which includes 13 active site residues, some of which display higher order epistasis). I agree with the idea that a surrogate fitness function (I like this wording much better than oracle, btw) is a useful strategy to compare sampling models at different temperatures in the absence of expansive (and expensive) experiments. This suggests an interesting strategy for other authors in future work, and I like the idea that this could be used to focus experiments around models and temperatures that are most likely to be successful or informative.

Thank you for your feedback. We have now appended the discussion to explicitly refer to this idea.

As a very minor comment, it appears that “surrogate” was swapped out for “oracle” with a search and replace; as a consequence in some spots the articles don’t match up (“an surrogate fitness function”).

Thank you, we have now corrected this mistake.

My only remaining request regards the code and software submission checklist, which seems incompletely satisfied.

It is great that the authors offer data and source code on git hub. The github repo seems reasonably well organized, however after poking around a little, my sense is that I would have a hard time getting started with it myself despite being a reasonably seasoned coder. More specifically:

- I don’t see a clear demo or use instructions – the README table of contents states that there is code to run on the manuscript data BUT not where it is or what to run. It would be good to add a top level file that outlines the general workflow for how to run the code on the data to obtain results in the paper or otherwise demo the code. To put it another way, if I wanted to try out the code on a new protein or data set, where would I start, and what order would I need to call the scripts in? Something as simple as knowing the order to do things in and paths to find the relevant files would be a big help.

Thank you for your comment. We have now updated the top level README file to point specifically to each example notebook which illustrates how to use the methods in the codebase to perform a particular analysis, such as inferring residue preferences from structural surrounding, or using such residue preferences for sampling combinatorial variants.

- I also don’t see the requested information on “expected output” or “expected runtime” as specified on the software submission checklist. It would be good to know what outputs should appear in what directories if running the code is successful. While I fully understand that specifying general runtimes is near impossible (as it depends on

machine architecture/environment), it would be useful to say approximately how much time the code took to run in your hands given a particular compute environment (specifying number of processors/memory requirements/GPUs vs CPUs would help here).

Thank you for your feedback. In addition to the jupyter notebooks detailing the output of each function, such as inferred residue preferences, which can be optionally saved to disk, we have also performed more comprehensive description for the expected runtimes given our hardware environments within these notebooks for example use cases of each analysis.

To summarize: the github repo is already quite good and doesn't need a major overhaul, but providing a little bit more documentation would go a long way in increasing accessibility/usability

Reviewer #2 (Remarks to the Author):

The comments in my original review were largely addressed.

'oracle' still used in Figure 1, Figure 4 text

Thank you for catching this, we have changed this now.

Some figure subpanel references for Fig. 3 need to be updated with the new Fig. 3c insertion

Thank you for catching this, we have updated these now.

I still think the issue of potential bias in the surrogate fitness function being used for comparison of different design algorithms/tools is an important point that the authors responded to in the response letter but didn't add to any discussion or caveat section of the manuscript that I see. This still seems to me like something that should be more explicitly discussed in the text.

Thank you for your comment. We have now explicitly introduced this caveat in the discussion section (page 19).

Reviewer #1 (Remarks to the Author):

The revisions address all of my concerns. Thank you!

Reviewer #1 (Remarks on code availability):

While I have not reviewed the code in detail, looking over the git hub repository and skimming a few notebooks indicates that it is well organized, appropriately commented, and includes an adequate README.

Reviewer #2 (Remarks to the Author):

Revision is fine